# Two-Pore-Domain Potassium Channel TREK–1 Mediates Pulmonary Fibrosis through Macrophage M2 Polarization and by Direct Promotion of Fibroblast Differentiation

**DOI:** 10.3390/biomedicines11051279

**Published:** 2023-04-26

**Authors:** Yunna Zhang, Jiafeng Fu, Yang Han, Dandan Feng, Shaojie Yue, Yan Zhou, Ziqiang Luo

**Affiliations:** 1Department of Physiology, Xiangya School of Medicine, Central South University, Changsha 410013, China; 2Department of Pediatrics, Xiangya Hospital, Central South University, Changsha 410013, China; 3Hunan Key Laboratory of Organ Fibrosis, Changsha 410013, China

**Keywords:** lung fibrosis, TREK–1, M2 macrophage, fibroblast, myofibroblast, p38

## Abstract

Idiopathic pulmonary fibrosis (IPF) is a devastating disease characterized by myofibroblast proliferation and abnormal accumulation of extracellular matrix in the lungs. After lung injury, M2 macrophages mediate the pathogenesis of pulmonary fibrosis by secreting fibrotic cytokines that promote myofibroblast activation. The TWIK-related potassium channel (TREK–1, also known as KCNK2) is a K2P channel that is highly expressed in cardiac, lung, and other tissues; it worsens various tumors, such as ovarian cancer and prostate cancer, and mediates cardiac fibrosis. However, the role of TREK–1 in lung fibrosis remains unclear. This study aimed to examine the effects of TREK–1 on bleomycin (BLM)-induced lung fibrosis. The results show that TREK–1 knockdown, mediated by the adenovirus or pharmacological inhibition of TREK–1 with fluoxetine, resulted in diminished BLM-induced lung fibrosis. TREK–1 overexpression in macrophages remarkably increased the M2 phenotype, resulting in fibroblast activation. Furthermore, TREK–1 knockdown and fluoxetine administration directly reduced the differentiation of fibroblasts to myofibroblasts by inhibiting the focal adhesion kinase (FAK)/p38 mitogen-activated protein kinases (p38)/Yes-associated protein (YAP) signaling pathway. In conclusion, TREK–1 plays a central role in the pathogenesis of BLM-induced lung fibrosis, which serves as a theoretical basis for the inhibition of TREK–1 as a potential therapy protocol for lung fibrosis.

## 1. Introduction

Idiopathic pulmonary fibrosis (IPF) is a lung fibrosis disease that is characterized by fibroblast proliferation and extracellular matrix (ECM) remodeling, which results in irreversible distortion of the lung architecture. Additionally, IPF has an average survival period of approximately three years after diagnosis [1,2]. Except for pirfenidone and nintedanib, which exhibit moderate efficacy on disease progression, there is no known effective treatment for IPF, but lung transplantation improves survival rates of patients with IPF [3]. It is believed that in fibrosis, an exaggerated and uncontrolled healing response occurs whereby key initiating features include inflammatory cell influx and the release of profibrotic products [4]. Macrophages are critical for the development of bleomycin(BLM)-induced pulmonary fibrosis and have been shown to produce soluble mediators that stimulate and recruit local tissue fibroblasts to differentiate into myofibroblasts and directly enhance the survival and activation of myofibroblasts, thus facilitating wound contraction and closure and the synthesis of ECM components [5]. The pathogenesis of IPF is not completely understood, and current therapies are limited [6]. Understanding the heterogeneity of these diseases and elucidating the final common pathways of fibrogenesis are critical for the development of efficacious therapies for severe fibrosing lung diseases.

Two-pore-domain potassium ion (K^+^) (K2P) channel subunits are made up of four transmembrane segments and two pore-forming domains that are arranged in tandem and function as either homo- or heterodimeric channels [7]. TREK–1 (KCNK2) is a K2P channel that is highly expressed in fetal neurons, cardiac tissue, and lungs [8,9,10,11]. Other K2P channels, such as TWIK–2, are known to regulate NLRP3 inflammasome activation in macrophages and septic-induced lung injury [12]. TREK–1 worsens various tumors, such as in ovarian cancer and prostate cancer, by promoting tumor cell proliferation and inhibiting tumor cell apoptosis [13,14]. TREK–1 also mediates the pathophysiological process of myocardial infarction [15] and has been implicated in the development of cardiac hypertrophy, fibrosis, and heart failure [11,16]. TREK–1 is known to be associated with inflammatory function in hyperoxia [17] or influenza A-induced lung injury [18]. Our reanalysis of published human single-cell sequencing data GSE110147 showed an upregulation of TREK–1 expression in lung tissue of IPF patients. However, the impact of TREK channels on lung fibrosis is unknown.

The current study aimed to evaluate the effect of TREK–1 on BLM-induced pulmonary fibrosis and specifically evaluate its mechanism of participation in fibrosis progression and effect on macrophages and fibroblasts.

## 2. Results

### 2.1. TREK–1 Expression Is Increased in the Lung Tissue of BLM-Challenged Mice

Before embarking on the role of TREK–1 in fibrogenesis, we characterized TREK–1 expression in fibrotic lung tissues. Mice with lung fibrosis stimulated by BLM were generated, and lung samples were collected (Figure 1A). As shown by H&E staining and Masson’s staining, alveolar damage, structural destruction, and accumulation of fibrotic collagen in the alveolar parenchyma, as well as upregulation of messenger RNA (mRNA) expression of α-smooth muscle actin (α-SMA) and collagen 1 (Col 1) after the BLM challenge for 14 days, indicated that the pulmonary fibrosis model was successfully established (Figure 1B–D).

TREK–1 expression was measured by immunofluorescence staining and qPCR in the lung tissues of mice with BLM-induced pulmonary fibrosis. The mRNA and protein expression levels of TREK–1 were markedly elevated in fibrotic lungs compared to the normal control lungs (Figure 1E,F and Appendix A). These data collectively demonstrate that TREK–1 expression is significantly increased in fibrotic lungs.

### 2.2. Inhibition of TREK–1 in the Lungs Attenuated BLM-Induced Fibrosis in Mice

To investigate the effect of TREK–1 in the experimental tissue fibrosis model, the classical bleomycin model of pulmonary fibrosis was employed, and the TREK–1 inhibitor fluoxetine (FLX) was used. Histologically, the FLX-treated mice with fibrosis exhibited a significant reduction in alveolar septum thickening and structural destruction compared to the BLM-challenged mice (Figure 2A). Masson’s staining showed that the total collagen in the lungs injured by BLM was reduced by FLX administration (Figure 2A). Immunohistochemical staining demonstrated that the expression of α-SMA and Col 1 was significantly increased in the lungs of BLM-challenged mice and was reduced by FLX administration (Figure 2B). Bleomycin exposure enhanced the expression of profibrotic genes (α-SMA, Col 1, and FN), which were attenuated by FLX treatment (Figure 2C–E). These results indicate that TREK–1 inhibition attenuates lung fibrosis in BLM-challenged mice.

### 2.3. Knockdown of TREK–1 Attenuated Bleomycin-Induced Pulmonary Fibrosis

As TREK–1 was upregulated in the fibrotic lungs of BLM-challenged mice. To further test the function of TREK–1 in lung fibrosis, we used adenovirus-mediated expression of short hairpin RNA directed against TREK–1 (shTREK–1) or an unrelated gene, LacZ (shLacZ). Adenovirus-mediated TREK–1 knockdown reduced the induction of TREK–1 by 66% in vivo (Figure 3A and Appendix A). Notably, H&E staining showed that alveolar septum thickening and collapse induced by bleomycin were significantly improved in TREK–1 knockdown mice (Figure 3B). Masson’s staining also showed that collagen deposition was markedly reduced in TREK–1 knockdown mice (Figure 3B and Appendix A). The protein expression of the profibrotic factors α-SMA, Col 1, and fibronectin (FN) was also significantly decreased in the lungs of TREK–1 knockdown mice (Figure 3C). Moreover, qPCR and the immunohistochemical staining results show that the expression of FN or α-SMA, Col 1, and collagen 3 (Col 3) markedly increased in lung tissues of the BLM-challenged mice and was significantly reduced in the TREK–1 knockdown mice (Figure 3D–G). These results demonstrate that the knockdown of TREK–1 attenuated BLM-induced lung fibrosis.

### 2.4. TREK–1 Overexpression Induces Macrophage Polarized to the M2 Phenotype

In the BLM-induced pulmonary fibrosis model, inflammatory macrophages increased immediately and reached a peak level 3 days after BLM induction and slowly reduced until day 21 [19]. Directly activated M1 macrophages usually infiltrate during the early stages of pulmonary fibrosis and participate in the initiation and maintenance of the inflammatory processes. This proinflammatory subtype may switch to an alternatively activated M2 phenotype, which is more likely to be associated with wound healing, tissue remodeling, and repair [20]. To assess whether TREK–1 is involved in the progression of deterioration in lung fibrosis attributed to macrophages or not, the expression of TREK–1 in macrophages from BLM-injured lungs was assessed. Histological examination showed that in the nonfibrotic control lungs, TREK–1 was barely detectable in macrophages expressing F4/80, whereas a significant upregulation of TREK–1 in macrophages was observed when treated with BLM at days 7 and 14 (Figure 4A). Considering the distinct polarized phenotypes of macrophages, we investigated the effects of TREK–1 on the mouse macrophage cell line RAW264.7. The RAW264.7 cells were transfected with TREK–1 plasmid or a negative control vector. TREK–1 overexpression was confirmed by quantitative PCR (Figure 4B). To directly assess whether TREK–1 mediates phenotypic changes in macrophages, q-PCR was performed, and the results show that there were no significant differences in the expression levels of the M1 markers tumor necrosis factor alpha (TNFα) and interleukin-1-beta (IL1β) (Figure 4C,D). However, a marked upregulation of the M2 markers arginase 1(Arg1) and mannose receptor C-type 1 (MRC1, also known as CD206) in TREK–1 overexpression cells, was observed (Figure 4E–H). These results illustrate that TREK–1 overexpression induces macrophage polarization into the M2 phenotype.

### 2.5. Macrophage Overexpression of TREK–1 Promotes Lung Fibroblast Transdifferentiation to Myofibroblasts through TGF-β1 Pathway

It has been suggested that the polarization of alveolar macrophages toward a profibrotic or M2 phenotype contributes to the development of fibrosis by secreting profibrotic cytokine-activating fibroblasts [21]. The present study shows that TREK–1 mediates macrophage M2 polarization, and a coculture system of macrophages and fibroblasts was constructed to further elucidate the effect of TREK–1 of macrophages on the activation of fibroblasts (Figure 5A). Our results show that TREK–1-overexpressing macrophages significantly increased the expression of profibrotic factor genes than NC plasmid-transfected cells, such as C-C motif chemokine receptor 2 (CCR2) especially transforming growth factor-β1 (TGF-β1), which is known to induce differentiation of fibroblasts to myofibroblasts (Figure 5B,C). Activated myofibroblasts are characterized by contraction via α-SMA and the synthesis and deposition of ECM [22] (including collagens, fibronectin, vitronectin, and tenascins). Conditioned media from macrophages expressing TREK–1 induced an increase in the expression of α-SMA, Col 1, vimentin, and FN in fibroblasts (Figure 5D–G), indicating that fibroblasts cultured with TREK–1-overexpressing macrophages transdifferentiated into myofibroblasts. Altogether, these results suggest that TREK–1 significantly enhances the establishment of a profibrotic microenvironment, by specifically increasing the production of profibrotic factors by macrophages, eventually indirectly mediating fibroblast activation.

### 2.6. Knockdown and Inhibition of TREK–1 Suppress Signaling Pathways Downstream of TGF-β1

Myofibroblasts are of major importance in the development of pulmonary fibrosis. As TREK–1-overexpressing macrophages increased the TGF-β1, and TGF-β1 is responsible for fibroblast activation into myofibroblasts, the TREK–1 knockdown in fibroblasts has been shown to prevent the progression of cardiac fibrosis [11]. To investigate the direct effect of TREK–1 on fibroblast activation in lung fibrosis, the expression of TREK–1 in myofibroblasts from fibrotic lungs was detected using double immunofluorescence staining of α-SMA and TREK–1. The images show that TREK–1 was significantly upregulated in α-SMA^+^ myofibroblasts of BLM-challenged mice (Figure 6A). Additionally, qPCR demonstrated that TREK–1 was significantly upregulated in TGF-β1-treated fibroblasts (Figure 6B). To investigate the effects of TREK–1 on fibroblast differentiation, TREK knockdown fibroblasts were generated using specific siRNA. In TGF-β1-stimulated fibroblasts, immunoblotting showed that the induction of fibrotic genes (α-SMA, Col 1, and FN) in response to TGF-β1 was inhibited by TREK–1 knockdown (Figure 6C and Appendix A). We further verified the critical role of TREK–1 in the promotion of fibroblast activation. Overexpression of TREK–1 in fibroblasts increased the expression of fibrotic proteins (α-SMA, Col 1, and FN) in fibroblast (Figure 6D and Appendix A). Moreover, the inhibition of TREK–1 by FLX effectively blunted the induction of cell growth (Appendix A and Figure 6E) and α-SMA expression (Figure 6F,G and Appendix A) in TGF-β1-treated cells in a concentration-dependent manner. These results demonstrate that TREK–1 directly mediates fibroblast activation. Focal Adhesion Kinase (FAK) and p38 mitogen-activated protein kinases (p38) are key regulators of the TGF-β1 pathway. Our results show that FAK, p38, and phosphorylated p38 were upregulated following BLM challenge, while Adv-mediated TREK–1 knockdown diminished their expression in fibrotic lung tissue (Figure 6H and Appendix A). Concordantly, FLX inhibited the phosphorylation of p38 in the fibroblasts (Figure 6I). Yes-associated protein (YAP) is a transcription factor that is important for coordinating fibrogenesis [23] and activation of p38-YAP signaling, which contribute to myofibroblast heterogeneity. Phosphorylated YAP is degraded in the cytoplasm and cannot translocate into the nucleus to promote fibrosis. Our results show that FLX treatment increased YAP phosphorylation (Figure 6I and Appendix A). Together, these results demonstrate that TREK–1 knockdown and inhibition may alleviate fibrosis by preventing fibroblast activation via the inhibition of FAK/P38/YAP signaling.

### 2.7. TREK–1 Is Upregulated in Lung Tissue of IPF Patients

Our results show that TREK–1 deteriorated BLM-induced lung fibrosis in mice. Furthermore, the effect of TREK–1 in IPF was described by using public database GSE110147, including lung samples from 22 IPF patients undergoing lung transplantation and 11 normal lung tissues surrounding lung cancer resections. An analysis of datasets showed that the TREK–1 mRNA level was upregulated in total lung tissue from IPF patients compared to the lung tissue of the controls (Figure 7A,B). To further assess whether TREK–1 is related to the macrophage phenotypic and functional changes, we detected the correlation between TREK–1 and the M1 or M2 marker in IPF lungs. The results demonstrate an inverse correlation between TREK–1 and the M1 marker tumor necrosis factor α(TNFα), while a direct correlation between TREK–1 and the M2 marker CD206 (Figure 7C,D) suggests that TREK–1 may induce macrophage polarization to the M2 type in IPF. The fibrotic genes α-SMA and Col 1 are strongly associated with TREK–1 (Figure 7E,F). These results demonstrate that the changes of TREK–1 in lung tissue from IPF patients and the pathogenesis of fibrosis are consistent with our findings in experimental pulmonary fibrosis models.

## 3. Discussion

In the present study, we used BLM-induced lung fibrosis mouse models to investigate the impact of TREK–1 on the pathogenesis of pulmonary fibrosis. The lungs originated from mice with lung fibrosis following BLM-induced upregulation of TREK–1 expression and a reanalysis of publicly available datasets also demonstrating an upregulation of TREK–1 in IPF lungs. The knockdown or inhibition of TREK–1 protected mice from BLM-induced lung fibrosis and suppressed fibroblast activation, suggesting TREK–1 promoted the development of lung fibrosis. TREK–1 was significantly upregulated in macrophages from BLM-challenged mice, and TREK–1 overexpression in macrophage cell lines regulated macrophage M2 polarization and increased CD206, Arg1, and Ym-1 expression. Importantly, macrophage overexpression of TREK–1 exacerbated the profibrogenic factor CCR2, especially TGF-β1 secretion, which then activated fibroblasts to initiate and promote the fibrotic processes. Furthermore, our results show that TGF-β1-induced fibroblast differentiation was diminished by TREK–1 interference and inhibition. TREK–1 knockdown suppressed the activation of the p38 MAPK pathway in BLM-induced lung fibrosis and TGF-β1-induced fibroblast activation. Thus, our findings show that TREK–1 mediates BLM-induced pulmonary fibrosis by promoting macrophage polarization to the M2 phenotype, which secretes TGF-β1 to indirectly activate fibroblasts. Moreover, TREK–1 can also be directly involved in fibroblast activation by targeting the FAK/p38/YAP pathway (Figure 8).

TREK–1, named KCNK2 or K2P2.1, belongs to a large family of K2P channels containing 15 members that are grouped into six subfamilies. The K2P channels are the most recently discovered class of K^+^ channel. K2P channels, or two-pore-domain potassium channels, are tandems of four transmembrane segments containing a two-pore domain [24]. TREK–1 is a mechanically sensitive channel, and mechanical tension through the lipid bilayer of the cell membrane can effectively activate TREK–1 channels. Additionally, TREK–1 activity is modulated by arachidonic acid, temperature, and pH [25,26,27] and is inhibited by neurotransmitters, hormones, and pharmacological agents, such as the antidepressant fluoxetine [28]. TREK–1 affects hyperoxia-induced acute lung injury by altering the function of alveolar epithelial cells [17]. A study showed that the knockdown of TREK–1 in fibroblasts may protect mice from stress overload-induced cardiac fibrosis [11], implying that TREK–1 plays an important regulatory role in fibrotic diseases. However, their specific roles in pulmonary fibrosis have not yet been reported. Our results show that the knockdown and inhibition of TREK–1 protected against BLM-induced pulmonary fibrosis.

Macrophages undergo marked phenotypic and functional changes after lung injury and play critical roles in the initiation, maintenance, and resolution phases of tissue repair [5]. Macrophages can be classically activated into the M1 type to participate in inflammation or alternatively activated into the M2 type and secrete profibrotic factors including epidermal growth factor-like growth factor amphiregulin, interleukin-10 (IL-10), platelet-derived growth factor α, and TGF-β1 to promote fibroblast transliteration into myofibroblasts [5,29,30,31,32,33]. Myofibroblasts are highly contractile cells with a strong ability to synthesize and secrete ECM to repair lung damage and a cell phenotype profibrotic following maladaptive repair responses characterized by pathologic fibrotic scarring [34]. Our results show that TREK–1 was rarely expressed in macrophages of normal lung tissue but was significantly upregulated on days 7 and 14 after bleomycin stimulation. In vitro, the overexpression of TREK–1 in macrophages controls the differentiation of macrophages to the M2 phenotype, which is conducive to fibrosis, specifically manifested by the upregulated expression of M2-type macrophage markers CD206, Ym1, and Arg1. Moreover, IPF datasets also elucidated a direct correlation between TREK–1 and the M2 biomarker CD206, but an inverse correlation with the M1 biomarker TNF-α. Our results suggest that macrophages overexpressing TREK–1 promote the expression of CCR2 and TGF-β1. In addition, we also demonstrated that the supernatant of macrophages overexpressing TREK–1 can promote fibroblast transdifferentiating to myofibroblasts, as shown by the elevated expression of α-SMA, vimentin, Col 1, and Col 3. Consistently, among lung samples from normal controls and IPF patients, a direct correlation was observed between TREK–1 and the two representative fibrosis biomarkers α-SMA and Col 1. However, its mechanism needs to be further explored.

Our results also indicate that TREK–1 is expressed in a-SMA^+^ cells (myofibroblasts) and increases after BLM challenge. Overexpression of TREK–1 exacerbated the upregulation of the fibrotic markers α-SMA, Col 1, and fibronectin protein; in contrast, the interference of TREK–1 in fibroblasts inhibited the transdifferentiation of fibroblasts. Moreover, fluoxetine, an inhibitor of TREK–1, prevents TGF-β1-induced fibroblast proliferation and activation in a concentration-dependent manner. This finding suggests that TREK–1 directly mediates fibroblast activation. p38 MAPK is a class of evolutionarily conserved serine/threonine mitogen-activated protein kinases that affects a variety of intracellular responses, with well-recognized roles in inflammation, cell cycle regulation, cell death, development, differentiation, senescence, and tumorigenesis [35]. Recent studies have shown that p38 MAPK plays an important role in idiopathic pulmonary fibrosis [36], and p38 MAPK mediates TGF-β1-induced fibroblast transdifferentiation to myofibroblasts in an FAK-dependent manner [37]. In our study, TREK–1 knockdown significantly inhibited p38 MAPK phosphorylation in BLM-induced lung fibrosis and in myofibroblasts stimulus by TGF-β1. The knockdown of TREK–1 in mice significantly inhibited the protein levels of FAK in fibrotic lung tissue. Notably, YAP, which mediates myofibroblast transcriptional activity, acts downstream of p38 and is both necessary and sufficient for inducing myofibroblast gene expression [38]. YAP phosphorylation results in cytoplasmic sequestration, blocking its nuclear translocation and inhibiting fibroblast activation [39]. Indeed, our results indicate that YAP phosphorylation was increased by fluoxetine in fibroblasts. Consequently, our results suggest that TREK–1, through the FAK/p38/YAP pathway, promotes fibroblast transdifferentiation into myofibroblasts.

To the best of our knowledge, this is the first study to identify the pathogenic role of TREK–1 in BLM-induced lung fibrosis. On the one hand, TREK–1 indirectly actives fibroblast by inducing macrophage-alternative activation to the M2 phenotype, which produces profibrotic cytokines. On the other hand, TREK–1 directly promotes fibroblast transdifferentiation to myofibroblasts through the FAK/P38/YAP pathway. In conclusion, our study identified the role of TREK–1 in lung fibrosis and provided insights into a promising therapeutic target to prevent the fibrogenesis. However, because fluoxetine is not a specific inhibitor of TREK–1, specific TREK–1 inhibitors, sodium ions inhibitors, and calcium ion inhibitors are required to demonstrate whether particular TREK–1 inhibition protects from bleomycin pulmonary fibrosis. As for fluoxetine’s off-target effects, these could be avoided by using TREK–1 mutants. The effect of fluoxetine on other cells especially macrophage cells from pulmonary fibrosis also needs further attention. A deeper understanding of how the cellular and molecular mechanisms of TREK–1’s function mediate pulmonary fibrosis needs to be clarified in the future.

## 4. Materials and Methods

### 4.1. Animal Experiments

All animals were housed under specific pathogen-free (SPF) conditions with free access to food and water. Briefly, for the lung fibrosis model, male C57Bl/6j mice (8 weeks, 21–24 g) received an intratracheal instillation of a single-dose bleomycin (50 μL, 1.25 mg/kg, Nippon Kayaku, Tokyo, Japan) or saline at day 0. For TREK–1 knockdown experiment, recombinant adenovirus carrying shRNA targeting mouse TREK–1 (Ad-shTREK–1: 5′-GCGTGGAGATCTACGACAAGT-3′ [40]) or adenovirus β-galactosidase (Ad-β-gal; control) used at 1 × 10^8^ PFU per mouse was injected into the tail vein of mice 7 days before BLM or saline airway installation. For the fluoxetine experiment, mice were administered TREK–1 inhibitor fluoxetine (FLX; MCE, Shanghai, China) ahead of bleomycin intratracheally at day 0, one receiving 100 μL fluoxetine (100× was dissolved in DMSO and diluted to working solution in normal saline) for 14 days (5 mg/kg once daily i.p.) and the other the vehicle control (1% DMSO). Each group included 6 mice for experiments.

### 4.2. Histological Assessment of Pulmonary Fibrosis

Lung tissues were fixed in 4% paraformaldehyde in phosphate buffered saline (PBS), embedded in paraffin. Hematoxylin and eosin (H&E) and Masson’s trichrome staining were performed using standard procedures.

### 4.3. Cell Culture

NIH 3T3 is a fibroblast cell line, and Raw264.7 is a macrophage cell line. All cell lines were purchased from Procell Life Science & Technology (Wuhan, China). These cells were maintained at 37 °C in 5% CO_2_ in Dulbecco’s modified eagle medium (DMEM; Procell, Wuhan, China) supplemented with 10% fetal bovine serum (FBS; Procell, Wuhan, China), 0.2% NaHCO_3_, 100 μg/mL streptomycin, and 100 IU/mL penicillin (Procell, Wuhan, China). NIH 3T3 cells were stimulated with 1, 10, and 20 μM fluoxetine or 10 ng/mL TGF-β1 (Sino Biological, Beijing, China) ranging from 5 min to 48 h according to specific experimental needs.

### 4.4. Plasmids and siRNA Transfections

Cells were grown to 70% confluence and transfected with 1 μg of the TREK–1 plasmid or control vector using lipofectamine 3000 (Invitrogen, Carlsbad, CA, USA) according to the manufacturer’s instructions. For RNA knockdown, NIH3T3 cells were transfected with 50 nM siRNA targeting TREK–1 or scrambled siRNA. For plasmid transfection of NIH3T3 and Raw264.7 cells, the plate was incubated at 37 °C for 6 h, and transfection complexes were removed and replaced with fresh media.

### 4.5. Preparation and Collection of Macrophage-Conditioned Medium

After macrophages were transfected with TREK–1 plasmid or negative control (NC) vector, transfection complexes were removed and washed three times with phosphate-buffered saline then replaced with fresh media and cultured for 48 h. The harvested conditioned medium was added to the fibroblasts separately. The cells were then incubated at 37 °C in a 5% CO_2_ incubator for 72 h.

### 4.6. RNA Extraction and q-PCR Experiments

Total RNA from lung tissue or cell samples were isolated using TRIzol reagent (Takara, Japan) and reversed-transcribed using a 5×TransScript^®^ Uni All-in-One SuperMix for qPCR (Transgene, Beijing, China) according to the manufacturer’s instructions. Real-time PCR was performed using PerfectStart^®^ Green qPCR SuperMix (Transgene, Beijing, China) and detected with a Bio-Rad real-time PCR system (CFX96 Touch™, Bio-Rad, Hercules, CA, USA). Relative gene expression was expressed as the relative fold of expression level in control group. The primer sequences are shown in Table 1.

### 4.7. Western Blotting

Total protein lysates were prepared in cold RIPA lysis buffer (Bioss, Beijing, China) containing a proteinase inhibitor cocktail (Apexbio, Houston, TX, USA) for 30 min. Lysates were clarified by centrifugation at 12,000× *g* for 10 min. A total of 20 μg protein per sample was separated by sodium dodecyl sulfate-polyacrylamide gel electrophoresis (SDS-PAGE), transferred to polyvinylidene fluoride (PVDF) membranes, and then probed with the indicated antibodies overnight at 4 °C. β-Actin or β-tubulin was used as loading control. HRP-conjugated anti-rabbit IgG (1:5000; Signalway Antibody, College Park, MD, USA) and HRP-conjugated anti-mouse IgG (H+L) (1:5000; Signalway Antibody, College Park, MD, USA) were used as secondary antibodies for 1 h at room temperature (RT). Subsequently, membranes were covered with enhanced chemiluminescence reagents (Cwbio, Taizhou, China) and imaged immediately using ChemiDoc XRS + (Bio-Rad Laboratories, Hercules, CA, USA). The primary antibodies were as follows: anti-β-actin (42 kd, 1:3000; Proteintech, Wuhan, China), anti-α-smooth muscle actin (α-SMA) (42 kd, 1:1000; Servicebio, Wuhan, China), anti-collagen I (Col 1) (130 kd, 1:1000; Proteintech, Wuhan, China), anti-Fibronectin (FN) (250 kd, 1:1000; Proteintech, Wuhan, China), anti-FAK (110 kd, 1:1000; Cell Signaling Technology, Danvers, MA, USA), anti-p-p38 (38 kd, 1:100; Abcam, Cambridge, UK ), anti-p38 (38 kd, 1:500; Santa Cruz Technology, Dallas, TX, USA), anti-p-YAP (70 kd, 1:3000; Affinity, Liyang, China), anti-YAP (70 kd, 1:3000; Proteintech, Wuhan, China), and anti-TREK–1 (47 kd, 1:1000; Alomone Labs, Jerusalem, Israel).

### 4.8. Immunofluorescence Staining

Lung tissues were fixed with 4% paraformaldehyde solution and embedded in paraffin. Then, 5 µm thick sections were deparaffinized in serial solutions of xylene, gradient ethanol, and water, followed by antigen retrieval by steaming in 1 mM EDTA for 5 min in a pressure cooker. The lung sections were washed with PBS buffer and blocked with 5% BSA for 30 min. Next, blocked samples were incubated with mouse anti-α-SMA (1:200; Servicebio, Wuhan, China) or anti-TREK–1 antibody (1:200, Alomone Labs, Jerusalem, Israel) and F4/80 antibody (1:200, proteintech, Wuhan, China) overnight at 4 °C followed by exposure to staining with corresponding secondary antibodies for 1 h at room temperature. Slides were then counterstained with nuclear dye DAPI and mounted. Images were acquired on a fluorescence microscopic imaging system (Leica, Weztlar, Germany).

### 4.9. Immunohistochemistry

After paraffin tissue sections were deparaffinized and repaired by pressure cooker, they were then incubated with 3% hydrogen peroxide for 10 min to inhibit endogenous peroxidase activity. The sections were stained with primary antibodies against collagen 1 (1:200, proteintech, Wuhan, China), α-SMA (1:200, proteintech, Wuhan, China), or collagen 3 (1:200, proteintech, Wuhan, China) overnight at 4 °C, followed by incubation with an HRP-conjugated secondary antibody (1:100; Sigma-Aldrich, Saint Louis, MO, USA) at room temperature for 30 min. DAB staining was implemented, and hematoxylin was used to indicate nucleus.

### 4.10. Human Microarray Data

GSE110147 was performed and analyzed according to the previous study [41]. Lung samples were from 22 IPF patients undergoing lung transplantation and 11 normal lung tissues flanking lung cancer resections.

### 4.11. Statistical Analysis

All data are presented as means ± the standard deviations (SD). Statistical comparisons between the two groups were analyzed with unpaired Student’s *t*-test. Comparisons among multiple groups were analyzed with one-way ANOVA, followed by the Student–Newman–Keuls test using GraphPad Prism software (GraphPad Software, San Diego, CA, USA), and the difference between groups was significant if *p* < 0.05.

## Figures and Tables

**Figure 1 biomedicines-11-01279-f001:**
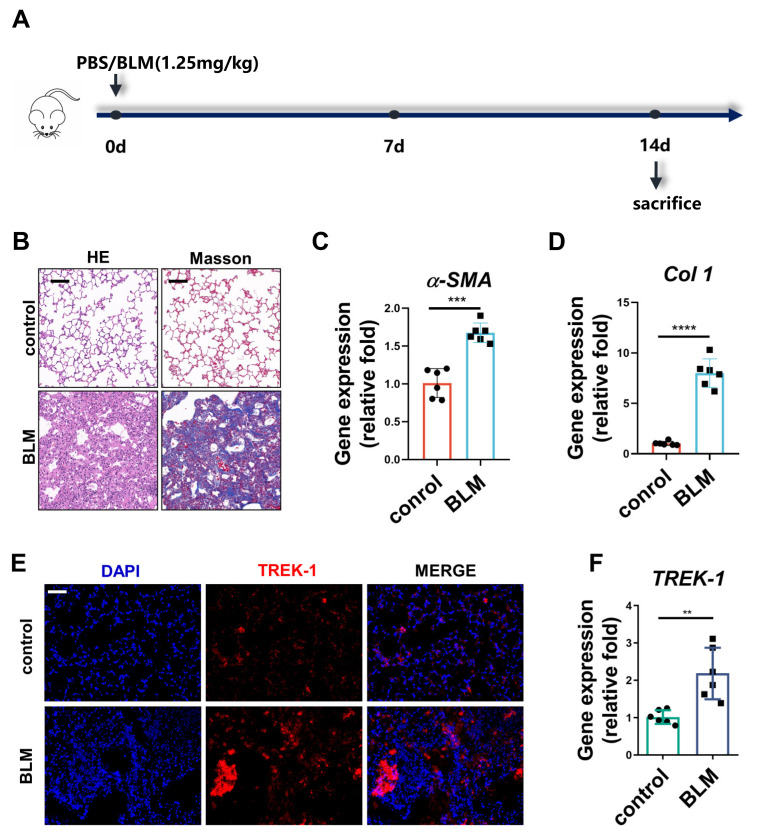
TREK–1 expression increased in murine lungs of BLM-induced lung fibrosis. (**A**) Schematic for BLM lung fibrosis model preparation procedure in mice. (**B**) Representative micrographs of H&E and Masson’s staining of lung sections, scale bar = 50 μm (*n* = 6). (**C**,**D**) mRNA levels of aSMA and Col 1 in the lung tissue of mice detected by qPCR (*n* = 6). (**E**) Immunofluorescent staining of TREK–1 in lung tissue of mice at 14 days after the BLM injection, scale bar = 50 μm (*n* = 3). (**F**) mRNA level of TREK–1 in the lung tissue of mice detected by qPCR (*n* = 6). ** *p* < 0.01, *** *p* < 0.001, **** *p* < 0.0001.

**Figure 2 biomedicines-11-01279-f002:**
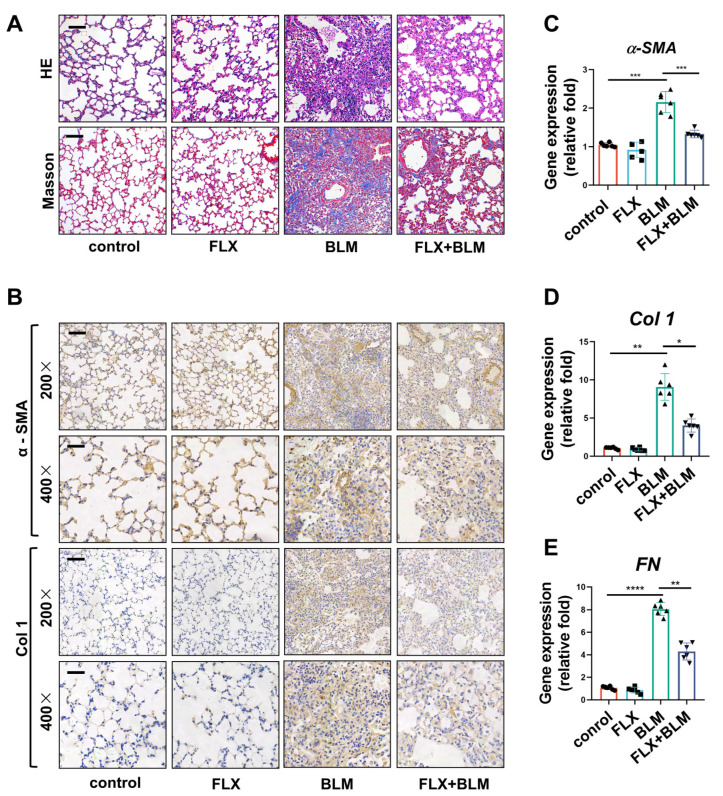
TREK–1 inhibition in the lungs attenuated by BLM-induced fibrosis in mice. (**A**) Representative micrographs of H&E and Masson’s staining of lung sections from indicated groups of mice, scale bar = 50 μm (*n* = 6). (**B**) Immunohistochemistry staining of α-SMA and Col 1 in the lung tissue of mice (magnification, ×200, bar = 50 μm and ×400, bar = 25 μm; *n* = 3). (**C**–**E**) mRNA levels of α-SMA, Col 1, and FN in the lung tissue of mice detected by qPCR (*n* = 6). * *p* < 0.05, ** *p* < 0.01, *** *p* < 0.001, **** *p* < 0.0001.

**Figure 3 biomedicines-11-01279-f003:**
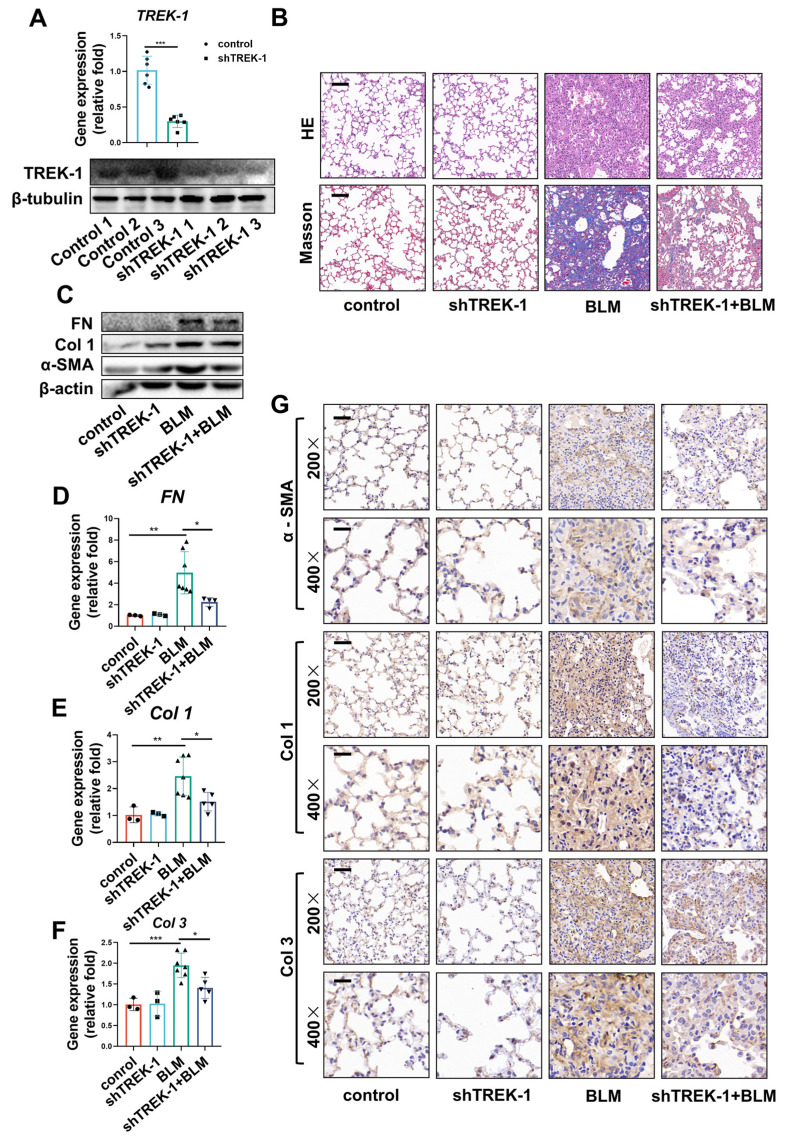
Knockdown of TREK–1 attenuated bleomycin-induced pulmonary fibrosis. (**A**) mRNA and protein levels of TREK–1 in the lung tissue from Ad-β-gal and Ad-shTREK–1 mice detected by qPCR, and Western blotting (*n* = 6). (**B**) Representative micrographs of H&E and Masson’s staining of lung sections from indicated groups of mice, scale bar = 50 μm (*n* = 3). (**C**) Contents of FN, Col 1, and α-SMA in the lung tissue detected by Western blotting; β-actin was used as the loading control (*n* = 3). (**D**–**F**) mRNA levels of FN, Col 1, and Col 3 in the lung tissue detected by qPCR (*n* ≥ 3). (**G**) Immunohistochemistry staining of α-SMA, Col 1, and Col 3 in the lung tissue of mice (magnification, ×200, bar = 50 μm and ×400, bar = 25 μm; *n* = 3). * *p* < 0.05, ** *p* < 0.01, *** *p* < 0.001.

**Figure 4 biomedicines-11-01279-f004:**
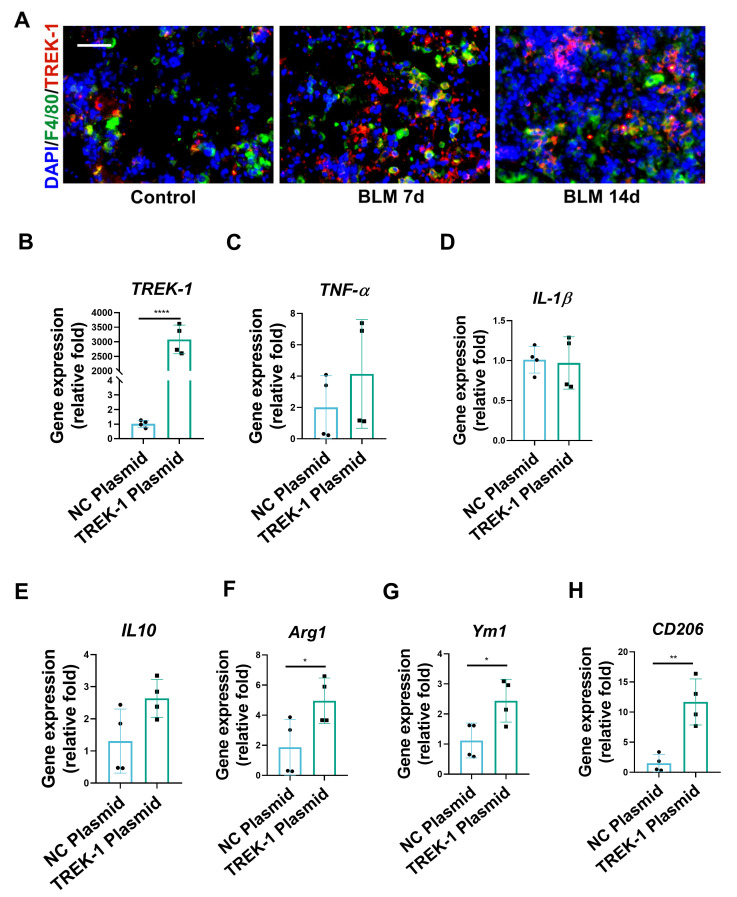
TREK–1 overexpression induces macrophages to polarize to the M2 phenotype. (**A**) Immunofluorescent double staining of F4/80 (Green) and TREK–1 (Red) of mice from control or BLM-injured lung sections, scale bar = 50 μm (*n* = 3). (**B**) mRNA level of TREK–1 in macrophages transfected with negative control (NC) or TREK–1 plasmid detected by qPCR (*n* = 4). (**C**,**D**) mRNA levels of M1 macrophage markers TNFa and IL-1β detected by qPCR (*n* = 4). (**E**–**H**) mRNA levels of M2 macrophage markers IL-10, Arg1, Ym1, and CD206 detected by qPCR (*n* = 4). * *p* < 0.05, ** *p* < 0.01, **** *p* < 0.0001.

**Figure 5 biomedicines-11-01279-f005:**
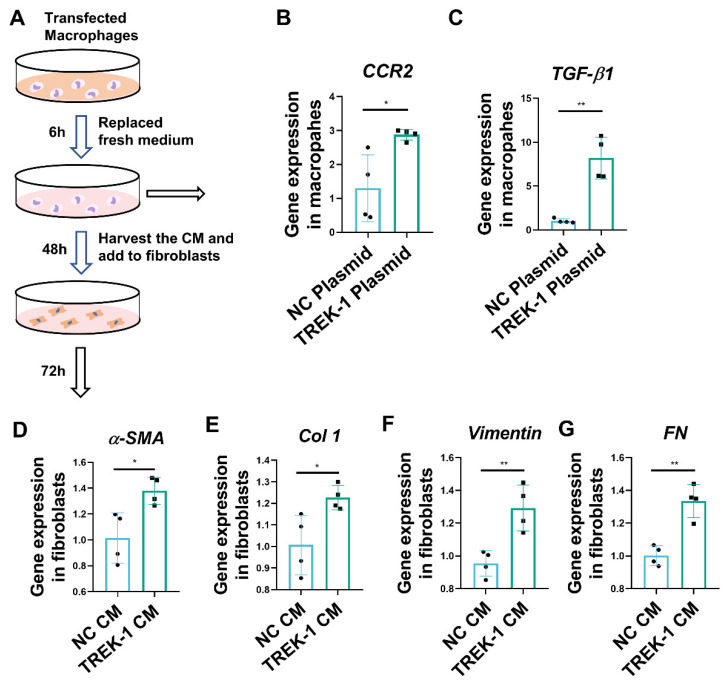
Macrophage overexpression of TREK–1 promotes lung fibroblasts transdifferentiation to myofibroblasts. (**A**) Experimental design: conditioned medium of NC or TREK–1 plasmid-transfected macrophages to culture fibroblasts. (**B**,**C**) mRNA levels of CCR2 and TGF-β1 in macrophages transfected with negative control (NC) or TREK–1 plasmid detected by q-PCR (*n* = 4). (**D**–**G**) mRNA levels of fibrotic genes α-SMA, Col 1, vimentin, and FN in fibroblasts detected by q-PCR (*n* = 4). * *p* < 0.05, ** *p* < 0.01.

**Figure 6 biomedicines-11-01279-f006:**
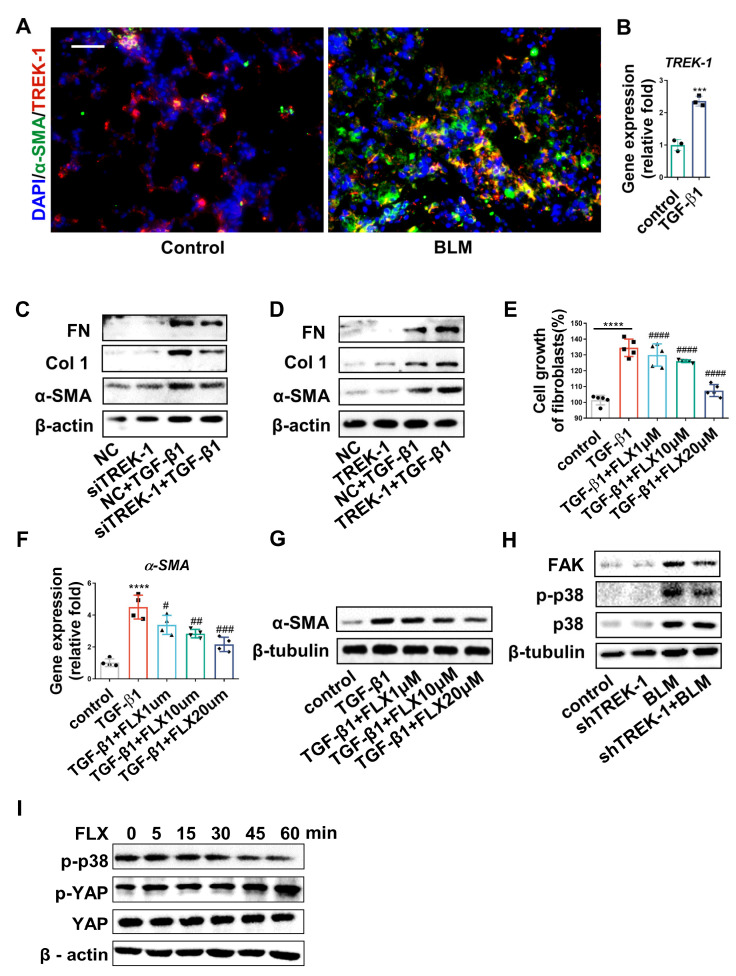
TREK–1 knockdown and inhibition suppress signaling pathways downstream of TGF-β1. (**A**) Immunofluorescent double staining of α-SMA (Green) and TREK–1 (Red) of mice from control or BLM-injured lung sections, scale bar = 50 μm (*n* = 3). (**B**) mRNA level of TREK–1 in fibroblasts detected by qPCR (*n* = 3). (**C**) Contents of α-SMA, Col 1, and FN in fibroblasts transfected with scrambled siRNA or siTREK–1 detected by Western blotting; β-actin was used as the loading control (*n* = 3). (**D**) Contents of α-SMA, Col 1, and FN in fibroblasts transfected with NC or TREK–1 plasmid detected by Western blotting; β-actin was used as the loading control (*n* = 3). (**E**) Growth of fibroblasts, evaluated by the CCK-8 assay. (**F**) mRNA level of α-SMA treated with vehicle control; TGF-β1 and fluoxetine detected by qPCR. (**G**) Contents of α-SMA in fibroblasts treated with vehicle control, TGF-β1, and fluoxetine detected by Western blotting; β-tubulin was used as the loading control. (**H**) Contents of FAK, p-p38 (phosphorylation p38), and total p38 in lung tissue detected by Western blotting; β-tubulin was used as the loading control. (**I**) Contents of p-p38, p-YAP, and total YAP in fibroblasts treated with vehicle control and fluoxetine detected by Western blotting; β-actin was used as the loading control *** *p* < 0.001, **** *p* < 0.0001 vs. control group. ^#^
*p* < 0.05, ^##^
*p* < 0.01, ^###^
*p* < 0.001, ^####^
*p* < 0.0001 vs. TGF-β1 group.

**Figure 7 biomedicines-11-01279-f007:**
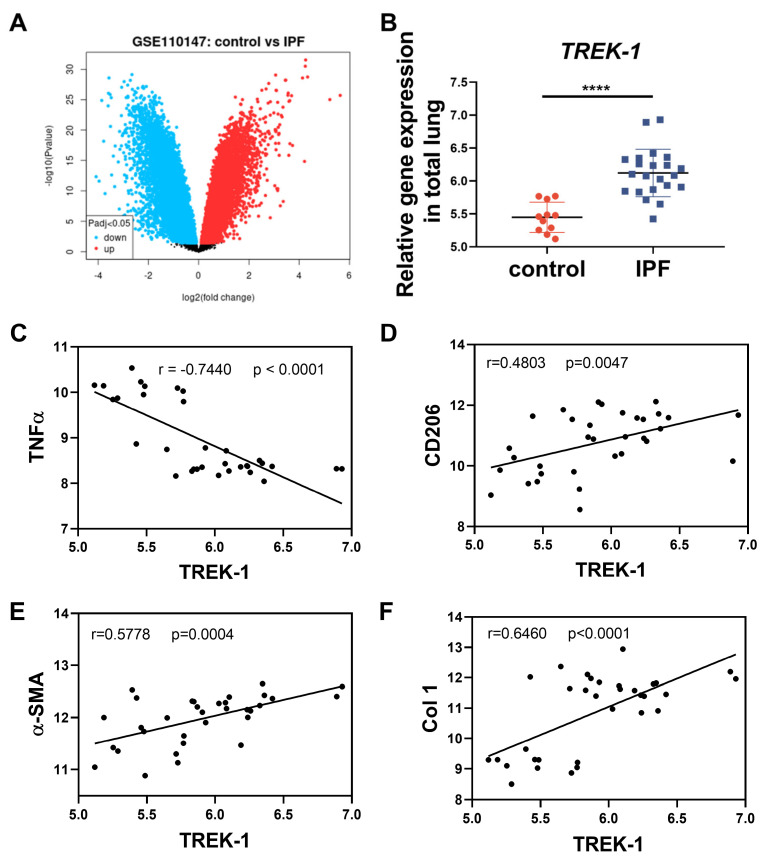
Analysis of publicly available microarray datasets showed TREK–1 is upregulated in lung tissue of IPF patients. (**A**) Volcano plot of differential genes between normal control and IPF. The upregulated genes are presented as red dots, and the downregulated genes are presented as blue dots. (**B**) Relative TREK–1 expression in total lung. (**C**,**D**) Correlation analysis between TREK–1 and macrophage biomarkers. (**E**,**F**) Correlation analysis between TREK–1 and fibrosis hallmarks. **** *p* < 0.0001.

**Figure 8 biomedicines-11-01279-f008:**
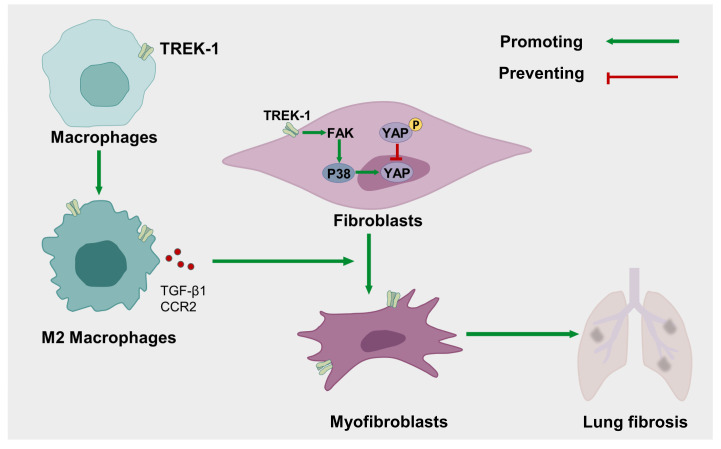
Summary of the effect of TREK–1 in lung fibrosis.

**Table 1 biomedicines-11-01279-t001:** Sequences of specific primers (mouse and human) used in this study.

	Sequence
*β-actin*	F: TTCCAGCCTTCCTTCTTG, R: GGAGCCAGAGCAGTAATC
*α-SMA*	F: CTTCGCTGGTGATGATGCTC, R: GTTGGTGATGATGCCGTGTT
*Col 1*	F: GAGCGGAGAGTACTGGATCG, R: GCTTCTTTTCCTTGGGGTTC
*Col 3*	F: GCACAGCAGTCCAACGTAGA, R: TCTCCAAATGGGATCTCTGG
*FN*	F: CCGACCAGAAGTTTGGGTTCT, R: CAATGCGGTACATGACCCCT
*TREK–1*	F: TTTCCTGGTGGTCGTCCTCTA, R: CTCGGTGGAGTTGACGCAG
*TGF-β1*	F: TTGCTTCAGCTCCACAGAGA, R: TGGTTGTAGAGGGCAAGGAC
*IL-1β*	F: GCCCATCCTCTGTGACTCAT, R: AGGCCACAGGTATTTTGTCG
*IL-10*	F: CCCATTCCTCGTCACGATCTC, R: TCAGACTGGTTTGGGATAGGTTT
*Arg1*	F: CTCCAAGCCAAAGTCCTTAGAG, R: AGGAGCTGTCATTAGGGACATC
*Ym1*	F: CAGCTCCTCTCAAAAGGATGTG, R: CTTGGGCAAACTGCTATCAGTAT
*CD206*	F: CTCTGTTCAGCTATTGGACGC, R: CGGAATTTCTGGGATTCAGCTTC
*CCR2*	F: ATCCACGGCATACTATCAACATC, R: CAAGGCTCACCATCATCGTAG
*Vimentin*	F: CGGCTGCGAGAGAAATTGC, R: CCACTTTCCGTTCAAGGTCAAG

## Data Availability

The datasets used and/or analyzed during the current study are available from the corresponding author on reasonable request.

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
