# Peer review of "Two-Pore-Domain Potassium Channel TREK–1 Mediates Pulmonary Fibrosis through Macrophage M2 Polarization and by Direct Promotion of Fibroblast Differentiation"

_biomedicines, 2023, doi:10.3390/biomedicines11051279_

Round 1

Reviewer 1 Report

Strengths:

1-     IPF is a clinically-relevant disease with limited treatment options.

2-     Combination of genetic and pharmacologic approaches.

3-     Combination of in vivo and in vitro approaches.

4-     Although K2P channels are fairly ubiquitously expressed, their role in the lung remains poorly understood.

5-     Mechanistic approach.

6-     Human IPF data.

Weaknesses/concerns:

1-     The current Introduction lacks a strong scientific rationale for studying TREK-1 in lung fibrosis, besides a brief comment on TREK-1 being involved in cardiac fibrosis. TREK-1 has known inflammatory functions in the lung, is upregulated in IPF in human sc-RNA datasets, other K2P channels like TWIK-2 are known to regulated macrophage/inflammasome biology, and how the role of TREK-1 in cardiac fibrosis may be relevant to the lung, could all be explained better to support a rational for the study.

2-     Fig. 1E would suggest almost absent expression of TREK-1 form control lungs. At what Ct level is TREK-1 picked up by PCR in Fig. 1F?

3-     To show TREK-1 expression in the lung is crucial for this study, but Fig.1 E does not provide much information without showing to which cell type(s) or lung structures TREK-1 localizes. Since the topic is IPF, showing co-localization of TREK-1 to fibrotic lung tissue would make this figure much more informative and valuable.

4-     Fig. 3A: Please provide a summary densitometry quantification for TREK-1 for what appears to be a western blot (not specified in the Legend). Similarly, please provide a summary densitometry quantification for Fig 3C and all western blots in Fig. 6.

5-     The strong signal for TREK-1 in control lungs in 3A (western blot) is not consistent with the virtual absence of TREK-1 in Fig. 1E (IF).

6-     TREK-1 expression in control lungs in 4A appears much stronger than in 1E. Overall, Fig. 4A appears overexposed. Can you turn down the intensity, and add an insert showing at higher magnification the expression pattern and membrane vs. cytosolic localization of TREK-1 on a macrophage? The cellular localization of TREK-1 on both macrophages and lungs is very important, since -as shown in Fig. 8 -the authors assume plasma membrane localization of TREK-1 on both macrophages and fibroblasts. Intracellular/cytoplasmic TREK-1 localization is known from other cell types, and may be responsible for the observed effects in this study. In fact, fluoxetine is highly membrane permeable (amphiphilic).

7-     Red and Green labeling between Fig 6A and the Legend do not align. Either way, however, there appears to be much less TREK-1 expression in this figure than in Fig 4A. Can you choose more consistent slides for all controls containing TREK-1?

8-     What are the units on the Y axis in Fig. 7B?

9-     Model/Fig 8: if fluoxetine is given to mice (or ultimately humans), wouldn’t it inhibit TREK-1 on both macrophages and fibroblasts? Why is this pathway only shown for fibroblasts?

10-  The discussion contains no section on limitations of the study. Just to point out one, fluoxetine is not a TREK-1-speciic inhibitor as it also inhibits voltage-gated Na as well as voltage-gated Ca channels. How could fluoxetine-off target effects interfere with the

11-  Please specify if the replicates indicated as “n” are indeed separate mice?

minor edits

Reviewer 2 Report

The authors of the study aimed to examine the effects of TREK-1 on bleomycin (BLM)-induced lung fibrosis. They found that TREK-1 plays a central role in the pathogenesis of bleomycin-induced lung fibrosis, in that TREK-1 knockdown resulted in diminished BLM–induced lung fibrosis and in reduced differentiation of fibroblasts to myofibroblasts. Moreover, TREK-1 overexpression in macrophages increased the M2 phenotype, resulting in fibroblast activation.

The manuscript is well written and very interesting.

Only few comments:

-Figure 3C: western blotting experiments as to be quantified

-Why did the authors choose to use male mice? Are gender differences involved in a different mechanism of development or progression of fibrosis?

-Details of the experimental protocol must be provided; how often is fluoxetine given? Does the protocol provide for a single administration of bleomycin? How long does it take for fibrosis to develop? It should take 14 days, but the authors show a 7-day image in Figure 4A.

-Figure 5. The legend is wrong. Please fix it

-Although human sample results are derived from a database, it is important to provide patient information (e.g. gender, age, comorbidities, …)

-Western blotting images: molecular weight of the different targets, has to be shown and/or described in material and methods section; moreover, the protein marker must be shown, at least in the original figure file

-Line 351: “1.25 mg kg-1” I think there is a mistake in this

-Please specify both the BLM and Fluoxetine vehicles

-Line 366: “Samples from at least three different individuals per group were tested in all experiments”. This sentence needs to be reformulated

-In the material and methods subsection 4.4, specify which type of cells were transfected. Macrophages?

-In the material and methods subsection 4.7, the authors assert that: “β-Actin was used as a loading control”. This is not true for the figure 2A, 6G, 6H in which the loading protein is β-tubulin. Please fix it in the dedicated section.

Round 2

Reviewer 1 Report

The authors addressed my concerns and suggestions in a sufficient manner.

minor editing required.